# Measurement of the Burden of Road Injuries in Colombia, 1990–2021

**DOI:** 10.3390/ijerph22081201

**Published:** 2025-07-31

**Authors:** Doris Cardona-Arango, Jahir Alexander Gutiérrez-Ossa, Gino Montenegro-Martínez, Ángela María Segura-Cardona, Diana Isabel Muñoz-Rodríguez, Liliana Giraldo-Rodríguez, Marcela Agudelo-Botero

**Affiliations:** 1Independent Researcher, Universidad CES, Cl 10A #22-04, El Poblado, Medellín 05002, Colombia; 2Escuela de Graduados, Universidad CES, Cl 10A #22-04, El Poblado, Medellín 05002, Colombia; 3Instituto Nacional de Geriatría, Avenida Contreras Número 428, Colonia San Jerónimo Lídice, Alcaldía Magdalena Contreras, Mexico City 10200, Mexico; 4Centro de Investigación en Políticas, Población y Salud, Facultad de Medicina, Universidad Nacional Autónoma de México, Centro Cultural 20, Insurgentes Cuicuilco, Coyoacán, Mexico City 04510, Mexico

**Keywords:** burden of disease, road injuries, mortality, disability-adjusted life-years, Colombia

## Abstract

Aim: To analyze the burden of road injuries in Colombia from 1990 to 2021, disaggregated by sex, age groups, and road injury category. Methods: Observational study based on the Global Burden of Diseases, Injuries, and Risk Factors Study (GBD) 2021. National data on prevalence, incidence, mortality, years of life lost (YLL), years lived with disability (YLD), and disability-adjusted life-years (DALY) were obtained. Data are reported in years and age-standardized and age-specific rates per 100,000 inhabitants. A log-linear segmented regression model was employed to analyze trends in DALY rates of road injuries from 1990 to 2021. Results: From 1990 to 2021, the age-standardized prevalence and incidence rates (per 100,000 inhabitants) due to road injury decreased by −30.6% (95% UI: −34.3; −26.4) and −27.5% (95% UI: −30.7; −24.4), respectively. The age-standardized mortality rate trend of road injuries decreased by −40.6% (95% UI: −50.0; −31.0). Meanwhile, the age-standardized DALY rate decreased by −39.7% (95% UI: −47.9; −31.3) during the study period. In all indicators, men’s rates were higher than women’s. By road injury category, the age-standardized rates increased significantly for motorcyclist road injuries, particularly among men. Conclusions: Road injuries in Colombia have declined but remain significant, especially for young men. Motorcycle injuries show alarming increases in mortality and DALY rates.

## 1. Introduction

Road injuries are a growing problem worldwide, especially in countries with lower socioeconomic development, where about 90% of road deaths occur [1]. Globally, road injury age-standardized mortality rates have decreased by −33.2% (95% uncertainty intervals [UI]: −37.7; −28.3), and −55.2% (95% UI: −57.6; −52.5) in high-income countries, and −34.7 (95% UI: −40.5; 28.6) in middle-income countries from 1990 to 2021 [2]. Among external causes, road injuries were the leading cause of disability-adjusted life years (DALY), with an age-standardized rate of 809.0 (95% UI: 752.1; 866.7) per 100,000 inhabitants in 2021 [2].

Moreover, these injuries place a high burden on individuals, families, and society, not only because of the impact resulting from premature deaths but also because of the sequelae and disability resulting from these injuries [1]. Of the 65,143,646.7 DALY worldwide in 2021 due to this cause, 12.7% was due to disability [2]. Countries with high levels of road injuries are associated with unsafe road infrastructure and weak legislation [3,4,5,6]. Consequently, costs for emergency medical care and rehabilitation for these injuries could reach USD 1.797 trillion globally over 2015-30 globally [7]. However, a large part of these costs is borne by public health systems.

In Colombia, the cost of health care for road injuries was more than COP 250 billion in 2018 (approximately USD 625 million), with motorcyclists, cyclists, and pedestrians being the users with the highest risk of dying on the country’s roads [8]. A growing motorization is observed in Colombia, causing urban traffic congestion, similar to what is happening in almost all of Latin America [9]. This phenomenon occurs mainly in the medium and low socioeconomic status populations, who opt for massive access to automobiles or motorcycles instead of using public transportation [9]. In addition, these vehicles are an essential source of livelihood in some of the poorest regions of the country, where motorcycles are used as a means of local transportation (known as motorcycle taxis) [5].

According to data from the Ministry of Transport, vehicle registrations (435,922) increased 75% in 2020, compared to 2019 [10]. Of this number, 73% were motorcycles, 11% were automobiles, and the rest were other vehicles [10]. However, this increase in motor vehicles is more significant than the modernization and adaptation of road and pedestrian infrastructure [9], and the regulatory measures implemented seem insufficient.

The situation described above suggests that road injuries will continue to be a significant public health problem in the country. Therefore, a broad epidemiological overview reflecting the impact on the population is necessary to serve as evidence for designing articulated and focused strategies and public policies [11]. This study analyzes the burden of road injuries in Colombia from 1990 to 2021, disaggregated by sex, age group, and road injury category.

## 2. Materials and Methods

### 2.1. Source of Information

An observational analysis [12] based on the Global Burden of Diseases, Injuries, and Risk Factors Study (GBD) 2021 by the Institute for Health Metrics and Evaluation (IHME) at the University of Washington was conducted [13,14]. This study aims to estimate the health losses caused by diseases, injuries, and risk factors at the global, regional, national, and local levels (the latter is unavailable for Colombia). It also provides a standardized analytical approach that estimates incidence, prevalence, mortality, years of life lost due to premature death, years lived with disability, and years of healthy life lost, among others. In 2021, the study covered 204 countries and territories and 371 diseases and risk factors from 1990 to 2021 [13,14]. The analysis of the GBD is based on various sources of information from censuses, surveys, hospital records, administrative records, and verbal autopsies [13,14]. The GBD also uses the Guidelines for Accurate and Transparent Health Estimates Reporting (GATHER) [13,14,15].

### 2.2. Case Definition and Classification of Road Injuries

A road injury refers to death or disability due to unintentional interaction with an automobile, motorcycle, bicycle, or other vehicle. The definition and classification of road injuries in the GBD study was based on the International Statistical Classification of Diseases and Related Health Problems 10th Revision (ICD-10) (V01–V04.9, V06–V80.9, V82–V82.9, V87.2–V87.3) [16,17,18]. For this study, road injuries were classified into five mutually exclusive categories of road injuries: (1) pedestrian road injuries (V01–V04.9, V06–V09.9); (2) cyclist road injuries (V10–V19.9); (3) motorcyclist road injuries (V20–V29.9); (4) motor vehicle road injuries (V30–V79.9, V87.2–V87.3); (5) other road injuries (e.g., animal riders, occupants of animal-drawn vehicles, and occupants of agricultural or special vehicles) (V80–V80.9, V82–V82.9) [16,17,18]. This study focuses exclusively on road injuries, as defined by the GBD framework (Level 3 cause). This category includes injuries from land transport incidents on public roads and excludes non-road or non-land transport injuries. This approach ensures alignment with GBD definitions and facilitates targeted analysis of road injuries.

### 2.3. Measurements

The GBD provided national data on prevalence, incidence, mortality, years of life lost (YLL), years lived with disability (YLD), and disability-adjusted life-years (DALY). Sex, five-year age groups, and categories of road injuries disaggregate the data.

Prevalence and incidence information are based on standard techniques to use available data and measure cause-specific epidemiological patterns. This method uses the Bayesian meta-regression tool DisMod-MR version 2.1 [13,14,19]. It includes information on the severity and occurrence of particular disease consequences to establish the proportion of prevalent cases experienced in each disease. Estimates can address the deficiencies often found in morbidity statistics, which rarely consider all cases occurring in a population or do not record the incidence of all health problems [13,14,19]. Using an iterative process, estimates can be reconciled to be consistent with the known epidemiological parameters of a population’s health problem [13,14,19].

The mortality rate was standardized directly, applying the global age structure of the World Health Organization [19]. The aforementioned is a weighted average of the age-specific rates for each population, which allows comparisons over time, regardless of the size and age structure of the population. YLL is a mortality measure calculated by multiplying the number of deaths from road injuries in each age group by the reference life expectancy at the average age of death of those who died in that age group. This indicator highlights premature deaths by applying a higher emphasis on deaths at younger ages. In turn, YLD was obtained by multiplying the prevalence of each cause by its disability weighting for each age, sex, or year. Total YLD are calculated by aggregating across all strata and sequelae. Disability weights are obtained from surveys. DALY results from the sum of the YLL and YLD of people with lower health status due to that cause [13,14,19].

### 2.4. Analysis

Data were disaggregated to the third and fourth levels of the GBD hierarchy, using ICD codes [20]. The reported data are in absolute numbers and age-standardized and age-specific rates. The percentage change of the indicators between 1990 and 2021 is presented. The 95% UI is reported for different measurements, represented by the 2·5th and 97·5th percentile values across the draws.

A joinpoint regression was performed to determine the trend in the age-standardized DALY rate. This analysis provides the segmented trend of the rate in periods and the amount of that change. Each point on the resulting line is determined by an annual percentage change (APC) based on the slope of the respective line, which is compared to the previous slope and shows whether the observed changes are statistically significant (α = 0.05) [21]. Also, the average annual percent change (AAPC) is reported [21], which provides information on the changes that occurred during the entire period (1990–2021). Analyses were completed using Stata^®^18 and the software Joinpoint Regression (Version 5.3.0) [22].

## 3. Results

### 3.1. Prevalence, Incidence, and Mortality Cases and Age-Standardized Rate of Road Injuries

From 1990 to 2021, the prevalence cases of road injuries in Colombia increased from 583,166.2 (95% UI: 543,978.9; 634,555.9) to 878,898.9 (95% UI: 812,557.4; 950,571.0), representing an increase of 50.7% (95% UI: 42.2; 60.5) in all periods (1990–2021); the number of incident cases was 300,230.1 (95% UI: 285,202.2; 316,275.3) in 1990 and 332,999.2 (95% UI: 314,598.9; 352,354.4) in 2021, which was an increase of 10.9% (95% UI: 5.8; 16.2) between 1990 and 2021. By sex, prevalence cases increased by 55.3% (95% UI: 45.1; 67.1) for men and 39.7% (95% UI: 33.7; 46.8) for women. The number of incident cases also showed an increase among men by 17.2% (95% UI: 10.3;25.1) and a reduction among women by −5.1% (95% UI: −9.8; −0.2).

Table 1 shows that age-standardized prevalence, incidence, and mortality rates (per 100,000 inhabitants) declined substantially between 1990 and 2021 for males and females across most injury categories. Notable exceptions to this trend were observed in motorcyclist road injuries, which showed substantial increases in prevalence, incidence, and mortality for both sexes. Additionally, mortality due to cyclist road injuries significantly increased among males.

### 3.2. YLD, YLL, and DALY Numbers and Age-Standardized Rate of Road Injuries

In 2021, a total of 50,836 (95% CI: 36,837.6; 67,292.3) YLD were recorded, representing a 34.1% (95% CI: 30.2; 48.4) increase compared to 1990, when the total was 37,901.4 (95% UI: 27,780.7; 49,929.8). The number of YLL decreased from 357,519.5 (95% UI: 342,636.5; 372,722.2) in 1990 to 351,189.9 (95% UI: 297,227.7; 406,347.2) in 2021, reflecting a −1.8% reduction (95% UI: −17.2; 13.6). The DALY count was 395,420.9 (95% UI: 376,645.2; 413,733.3) in 1990 and 402,025.9 (95% UI: 34,039.9; 462,954.2) in 2021, representing a 1.7% (95% UI: −12.1; 16.0) increase. The DALY number increased by 8.0% (95% UI: −6.9; 24.0) in men, while it decreased by −19.1% (95% UI: −30.3; −7.1) in women.

Table 2 shows the age-standardized rates (per 100,000 inhabitants), for all ages and both sexes, decreased for YLD, YLL, and DALY by −33.8% (95% UI: −41.0; −33.8), −30.5% (95% UI: −49.5; −30.5), and −39.7% (95% UI: −47.9; −31.3) from 1990 to 2021, respectively. It should be noted that the highest DALY burden of road injuries was due to YLL, which means that 87.6% of the age-standardized DALY rate in 2021 was due to premature deaths. By sex, it can be observed that the age-standardized rates for road injuries, across all measurements, were higher in men than in women for all causes of road injuries and all indicators analyzed. Furthermore, although these rates decreased for both men and women, the reductions were more pronounced in women. Regarding road injury category, the age-standardized DALY rate due to motorcyclist road injuries increased by 181.5% (95% UI: 143.4; 224.2): 187.3% (95% UI: 146.4; 236.1) for men and 134.1% (95% UI: 99.2; 174.8) for women.

In 2021, the percentage distribution of DALY from road injuries, disaggregated by sex, age group, and road injury category, showed the highest burden from pedestrian road injuries. This burden was most prominent among children aged 0–14 years (both sexes), women aged 50 years and older, and men aged 55 years and older. For men aged 15 to 54 years, the highest percentage of DALY was due to motorcycle road injuries; in contrast, women aged 15 to 44 years had the highest DALY loss due to the same cause. Similarly, DALY attributable to cyclist road injuries were highly concentrated in children aged 10–14 years; however, these deaths represented 19.0% of the total DALY of road injuries for boys and 8.4% for girls. It is also noteworthy that cyclist road injuries accounted for more than 10% of their respective DALY in men aged 60 years and older, and this value was notably higher for the age group of 65–69 years (14.0%). Finally, women of all ages had a higher percentage of DALY due to motor vehicle road injuries than men (Figure 1).

Figure 2 shows the age-specific DALY rate from road injuries for 1990 and 2021 and the percentage change in this rate between these years, disaggregated by sex and age groups. Overall, this rate decreased across all age groups, for both men and women. The most considerable reductions in the age-specific DALY rate of road injuries were observed among people aged 0 to 14 years of both sexes and women aged 60 years and older. Age-specific DALY rates were similar in the first two age groups; however, the gap between age groups and sexes became wider as age increased.

### 3.3. Joinpoint Regression Analysis of Age-Standardized DALY Rate of Road Injuries

Table 3 summarizes the results of the joinpoint regression analysis assessing changes in age-standardized DALY rates for road injuries. Overall, road injuries exhibited a statistically significant average annual decrease of 1.7%. A similar trend was observed for motor vehicle injuries and pedestrian road injuries, although the decreases were more pronounced for both causes when considering the overall burden. In contrast, the age-standardized DALY rate for motorcycle road injuries had an average annual increase of 3.4% (α = 0.05). By period, there were changes in the rate that were statistically significant, as well as very pronounced and heterogeneous. For example, road injuries decreased by 5.1% from 1996 to 2000. In motor vehicle road injuries, there were also two periods with rate decreases of more than 5%, while the pedestrian road injury rate had notable decreases in 1994–2000 and 2009–2013.

Regarding cyclist road injuries, there was a percentage increase that lasted until 2004 in the first two periods and then decreased in the last three periods analyzed. Unlike other road injury categories, age-standardized DALY rates for motorcyclist injuries increased across five periods—statistically significant in four—and declined only in the most recent period (2015–2021), with an average annual reduction of 3.4%.

## 4. Discussion

The impact and trend of road injuries in Colombia were analyzed using different sources of information, periods, and geographic areas [23,24]. Furthermore, this is the first analysis in the country using data from the GBD. This information is helpful as it provides a comprehensive view of the burden of road injuries through key indicators that measure, among other things, YLL, YLD, and DALY.

Despite the overall reduction in the age-standardized mortality rate for road injuries, these injuries remained among the ten leading causes of mortality in Colombia in 2021, ranking seventh among men. This finding underscores their persistent relevance within the country’s epidemiological landscape [2]. The observed decline is consistent with reductions reported in Central American countries, such as Nicaragua (−42.0%) and El Salvador (−39.8%). A similar downward trend in age-standardized mortality rates was documented at the global level over the study period. Most Latin American countries followed this pattern, with the exceptions of Guatemala and Paraguay, where mortality rates increased by 6.4% and 26.3% between 1990 and 2021 [25].

Unlike other causes, road injuries are highly preventable and avoidable [26]. Hence, measures are urgently needed to significantly decrease the mortality and disability associated with them. Latin America, Asia, and the Middle East have had some of the highest increases in motorization worldwide, which is alarming [9]. It is worth noting that the Sustainable Development Goals (SDGs) aim for a 50% reduction in deaths and injuries from road injuries by 2030 (SDG target 3.6) [27]. This will be achieved through access to safe, affordable, accessible, and sustainable transport systems for everyone by improving road safety (SDG target 11.2). Although Colombia has made some progress, it is still far from meeting these targets [23]. By 2031, it is hoped that Colombia will reduce the fatality rate of road injuries to 7.1 per 100,000 inhabitants [28].

Regarding regulations, various actions have been implemented in the country [28,29]; however, few studies account for the effects these measures have had on the health burden of road injuries [30]. In addition, the Ministry of Health and Social Protection prioritized addressing road safety as a public health problem, considering the number of deaths, injuries, and people with disabilities [31]. This regulatory framework has included, among other things, specific measures such as the use of seat belts, penalties for people who drive under the influence of alcohol or other psychoactive substances, the specification of speed limits for urban roads and highways, the use of security cameras, and the use of helmets for motorcyclists, which must meet specific safety requirements [29].

Despite all this, in the country, 50% of road injuries involve at least one injured or deceased person; of the latter, 70% of people die on the same day of the incident. This means that 44% of these injuries are considered severe. In 2018, the rate of YLL for the average Colombian population was 4.2 per 100,000 inhabitants: 1.3 for women and 6.8 for men. This indicator has remained almost the same since 2005 [24].

By road injury category, motorcycle road injuries were the only ones that had a consistent increase in all indicators of the burden of disease. In 2021, Colombia ranked sixth globally for the highest age-standardized DALY rate due to this cause and fourth in Latin America, behind Paraguay, Brazil, and Costa Rica [2]. In Colombia, 56% of the total vehicle fleet comprises motorcycles, which cause 53% of road fatalities annually [31]. According to Ospina-Mateus et al., the social cost of a life lost in a motorcycle road injury is approximately USD 725,400 per year [32]. The sustained increase in the burden of motorcycle road injuries in Colombia poses a critical challenge for public health and mobility policy. These injuries, which primarily affect young men of working age, lead to economic losses and place substantial pressure on emergency services, rehabilitation systems, and social protection mechanisms. There is an urgent need to implement comprehensive road safety strategies, including mandatory use of certified helmets, periodic technical inspections, traffic law enforcement, and targeted education campaigns [32,33,34].

On the other hand, pedestrian injuries and deaths are still very relevant in the profile of road injuries, which is why interventions urgently need to be strengthened. In particular, interventions that protect children and older adults are important because they are more affected than other age groups. This high susceptibility to death or disability has been explained by a greater vulnerability of children and older adults, which is consistent with other studies on the subject [35,36,37,38].

The burden of cyclist road injuries, which are becoming increasingly common, should also be highlighted. However, very little has been studied on this means of transportation in the country. In a recent study conducted in Bogotá (the capital city of Colombia), cyclist road injuries exhibited social, economic, and spatial inequalities, which are reflected in three aspects: (1) disparities in the allocation of infrastructure; (2) concentration of pedal cyclist deaths in low-income areas; and (3) disproportion in the coverage of these deaths, depending on their geographic location [39]. In addition, records of cyclist road injuries tend to be monocausal, i.e., they focus on variables inherent to the behavior of the pedal cyclist, leaving aside the characteristics of the environment, infrastructure conditions, and interactions with other road users [40].

The determinants of road injuries are multifactorial [24,41,42], among which speeding and failure to comply with transportation regulations stand out. In Colombia, a relevant factor is that many vehicles do not undergo a technical–mechanical inspection at least once a year, as stipulated in the rules [29]. It is known that 55% of the vehicle fleet does not have such an inspection; this percentage is higher for motorcycles (68%), followed by private vehicles (38%) and public service vehicles (24%) [23]. The high percentage of motorcycles circulating without this inspection in the country makes them a hazardous vehicle for road users. Added to this is the high evasion of compulsory vehicle insurance (which covers body damages caused to the people involved in the event, whether they are pedestrians, passengers, or drivers), estimated at 47.1%, 60.5% for motorcycles and 27.4% for motor vehicles [43].

This study also provides important insights into the disparities in the burden of transport-related injuries by sex and age group. The findings are consistent with global evidence showing that men and younger individuals consistently exhibit the highest mortality rates from road injuries compared to other age groups [25,44,45]. These differences have been partially attributed to increased motorization, a higher prevalence of risk-taking behaviors among men, and the limited enforcement of road safety measures specifically targeting certain demographic groups, among other factors [25,44].

The 2021 GBD analysis showed that 40% of deaths from road injuries occurred among individuals aged 15–49 years [25]. It also documented that mortality rates were higher in men than in women, including at older ages, where sex differences tend to narrow [25]. In the 15–49 age group alone, men had nearly double the age-standardized mortality rate compared to women (22.8 vs. 11.9 per 100,000 inhabitants) [25]. Moreover, men accounted for 60% of the total YLD in 2021 [25]. Although men are more frequently involved in road injuries, women often experience more serious health outcomes from accidents of similar severity [46].

Data from Colombia suggest the need to implement ad hoc strategies tailored to each age and sex group, given the wide variability in age-standardized YLD rates across population subgroups. However, current public policies lack a life-course and gender-sensitive approach [47]. Such policies should be cross-cutting and intersectoral, while also addressing specific risk exposures and contexts of vulnerability [41,47].

### Study Limitations

One of the limitations of this study is that data in the GBD are only available at the national level, so it is impossible to know the geographic differences regarding road injuries. In other studies, conducted in the country, essential contrasts were found in the occurrence and fatality of road injuries [23,24]. Therefore, it is highly recommended that a subnational perspective be incorporated into this type of analysis. Another limitation is the possible underreporting of mortality and disability cases, affecting the results. The GBD 2021 employs different techniques to solve the problems of collecting and measuring information to obtain reliable estimates. Other authors reported in 2018 that the national percentage of duly certified deaths was 93.7%, which places Colombia as a country with an acceptable quality of information on deaths [48]. Non-fatal injuries are likely underreported, particularly in low- and middle-income countries, where many cases go undocumented due to the heterogeneity and limited capacity of local institutions to collect timely and detailed data [25]. This highlights the critical need for accurate and coordinated health information systems.

## 5. Conclusions

Although road injuries have declined in Colombia over the past 32 years, they continue to represent a significant contributor to the national burden of disease, particularly among young people and men. However, disaggregation by road injury category reveals considerable heterogeneity in both the magnitude and direction of trends. Moreover, certain population subgroups remain especially vulnerable to specific types of injuries. Overall, the entire population—regardless of sex or age group—remains at risk of harm from transport-related incidents on public roads. Therefore, timely and comprehensive information is essential to equip decision-makers with the necessary tools to design, adapt, and implement effective public policies informed by a gender-sensitive and life-course approach.

Among the various categories, motorcycle-related injuries are particularly concerning, as mortality and DALY rates have increased over the study period. Addressing this issue requires policies that comprehensively consider the range of risks faced by road users. Interventions must be targeted, multisectoral, and, most importantly, sustained over time.

## Figures and Tables

**Figure 1 ijerph-22-01201-f001:**
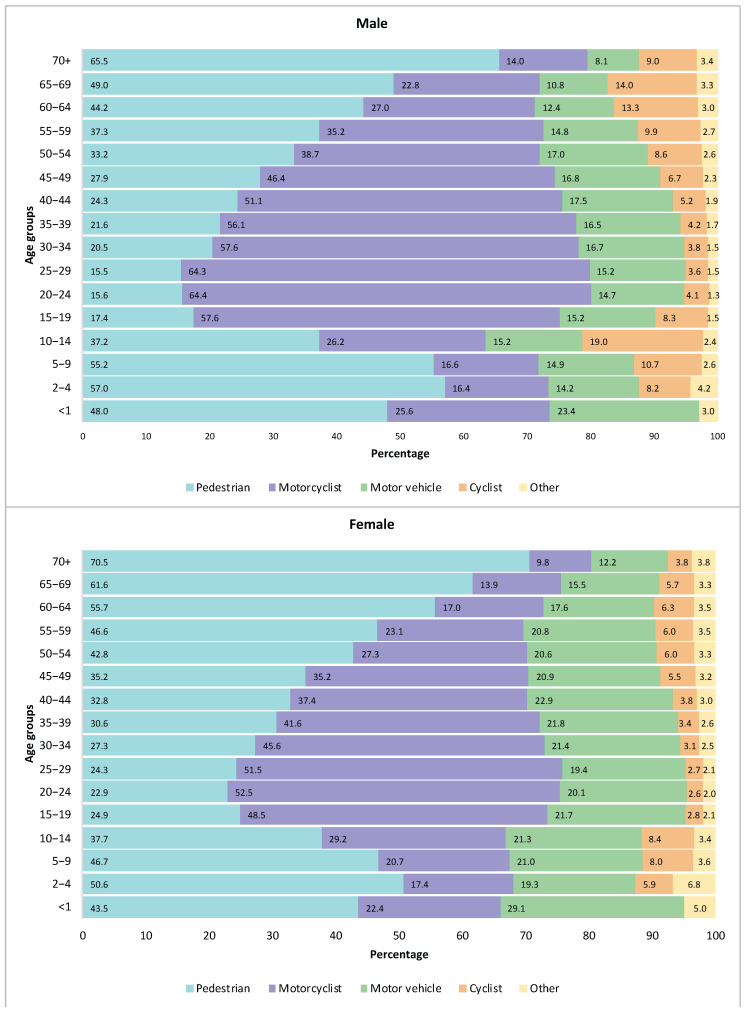
Percentage distribution of the number of DALY from road injuries by road injury category, according to sex and age groups. Colombia, 2021. Source: own elaboration based on GBD 2021.

**Figure 2 ijerph-22-01201-f002:**
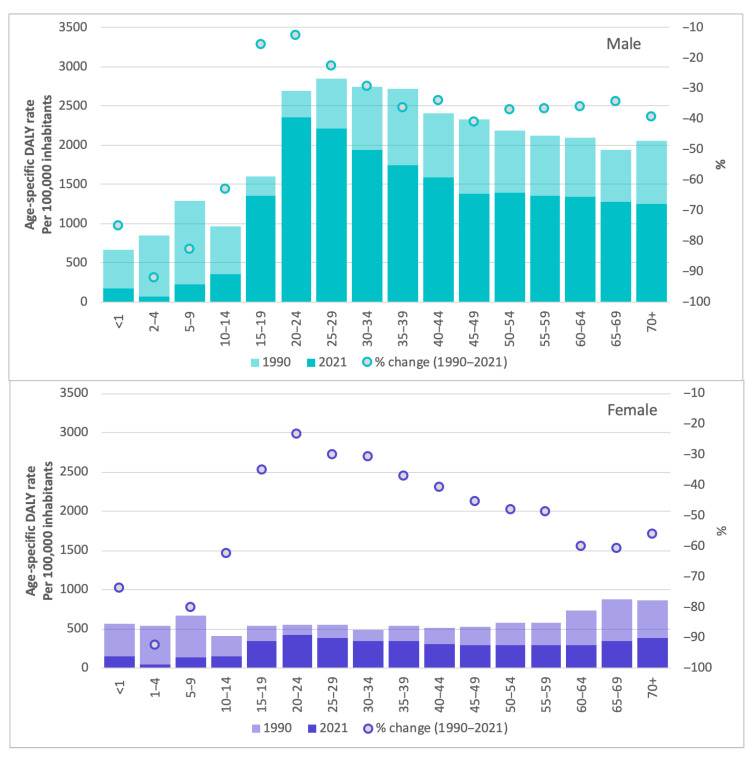
Age-specific DALY rates of road injuries by sex and age groups. Colombia, 1990 and 2021. Source: own elaboration based on GBD 2021.

**Table 1 ijerph-22-01201-t001:** Age-standardized prevalence, incidence, and mortality rates of road injuries. Colombia, 1990 and 2021.

**Cause**	**Age-Standardized Prevalence Rate**								
	**(95% Uncertainty Intervals)**								
	**Male**		**% Change**	**Female**		**% Change**	**Total**		**% Change**
	**1990**	**2021**	**1990–2021**	**1990**	**2021**	**1990–2021**	**1990**	**2021**	**1990–2021**
Road injuries	3418.8	2499.8	−26.9	628.9	854.0	−35.7	2346.6	1629.0	−30.6
	(3200.0; 3730.3)	(2294.1; 2722.3)	(−31.6; −21.6)	(566.7; 705.0)	(803.1; 900.1)	(−38.1; −32.7)	(2198.2; 2542.0)	(1506.1; 1760.8)	(−34.3; −26.4)
Motor vehicle road injuries	709.7	298.5	−57.9	322.4	163.0	−49.4	511.3	226.7	−55.7
	(618.5; 815.8)	(260.9; 342.5)	(−59.6; −56.5)	(282.8; 363.2)	(143.4; 183.1)	(−51.4; −47.8)	(448.8; 583.7)	(199.1; 257.2)	(−57.2; −54.5)
Pedestrian road injuries	1697.1	445.2	−73.8	628.9	190.6	−69.7	1148.9	308.7	−73.1
	(1515.4; 1905.9)	(397.0; 510.1)	(−74.9; −72.6)	(566.7; 705.0)	(172.8; 211.1)	(−70.9; −68.3)	(1030.8; 1287.1)	(277.9; 348.5)	(−74.2; −72.1)
Motorcyclist road injuries	308.8	1207.5	291.0	107.7	282.6	162.5	205.8	719.5	249.6
	(263.1; 355.3)	(1023.9; 1395.9)	(270.1; 313.3)	(94.4; 122.3)	(244.6; 320.1)	(144.3; 179.4)	(177.4; 233.7)	(613.3; 826.7)	(230.0; 268.8)
Cyclist road injuries	407.4	399.0	−2.1	137.5	124.7	−9.3	269.2	254.1	−5.6
	(346.7; 469.9)	(336.6; 459.9)	(−7.2; 2.7)	(114.9; 164.5)	(106.3; 146.8)	(−13.6; −4.9)	(231.0; 310.7)	(217.8; 293.0)	(−10.0; −1.5)
Other road injuries	364.9	188.5	−48.3	140.4	97.4	−30.7	249.9	140.3	−43.9
	(317.4; 424.8)	(166.4; 217.9)	(−50.2; −46.2)	(121.0; 162.5)	(84.4; 110.9)	(−33.8; −28.0)	(217.9; 290.0)	(124.4; 160.7)	(−45.9; −41.9)
**Cause**	**Age-Standardized Incidence Rate**								
	**(95% Uncertainty Intervals)**								
	**Male**		**% Change**	**Female**		**% Change**	**Total**		**% Change**
	**1990**	**2021**	**1990–2021**	**1990**	**2021**	**1990–2021**	**1990**	**2021**	**1990–2021**
Road injuries	1324.1	982.8	−25.8	502.5	332.4	−33.8	903.5	654.6	−27.5
	(1254.7; 1390.7)	(923.6; 1045.8)	(−29.9; −21.1)	(476.2; 532.8)	(312.6; 355.5)	(−36.9; −30.6)	(862.9; 947.4)	(618.3; 690.3)	(−30.7; −24.4)
Motor vehicle road injuries	338.7	160.8	−52.5	155.6	86.6	−44.4	244.5	123.3	−49.6
	(309.0; 370.2)	(143.1; 179.4)	(−56.4; −48.5)	(136.6; 176.3)	(73.7; 98.7)	(−48.3; −39.6)	(220.6; 268.5)	(109.3; 136.3)	(−52.7; −46.4)
Pedestrian road injuries	561.3	166.7	−70.3	206.3	69.2	−66.5	379.3	116.4	−69.3
	(516.9; 601.8)	(151.6; 180.3)	(−72.4; −68.1)	(188.9; 224.8)	(61.9; 77.0)	(−68.9; −63.8)	(352.4; 406.1)	(107.0; 126.6)	(−71.0; −67.5)
Motorcyclist road injuries	75.6	363.2	380.6	24.7	77.6	214.1	49.4	220.2	345.2
	(65.9; 85.3)	(327.2; 400.0)	(332.6; 431.5)	(19.3; 30.3)	(67.8; 88.0)	(171.6; 275.7)	(43.3; 56.2)	(199.8; 242.0)	(301.8; 394.1)
Cyclist road injuries	192.4	206	7.1	53.7	52.9	−1.5	121.8	128.8	5.7
	(160.1; 233.6)	(181.7; 237.6)	(−4.2; 21.4)	(39.9; 74.4)	(40.3; 71.0)	(−11.7; 10.1)	(100.3; 151.0)	(112.5; 150.3)	(−4.1; 18.3)
Other road injuries	156.1	86.1	−44.9	62.1	46.1	−25.8	108.3	65.8	−39.2
	(123.0; 194.8)	(68.9; 104.3)	(−50.2; −39.2)	(47.3; 78.2)	(35.7; 56.3)	(−32.6; −18.4)	(85.8; 134.1)	(52.6; 79.1)	(−43.9; −34.4)
**Cause**	**Age-Standardized Mortality Rate**								
	**(95% Uncertainty Intervals)**								
	**Male**		**% Change**	**Female**		**% Change**	**Total**		**% Change**
	**1990**	**2021**	**1990–2021**	**1990**	**2021**	**1990–2021**	**1990**	**2021**	**1990–2021**
Road injuries	37.0	23.6	−36.3	10.8	5.1	−52.7	23.5	14.0	−40.6
	(35.7; 38.3)	(19.7; 27.5)	(−46.7; −25.1)	(10.2; 11.3)	(4.3; 5.9)	(−60.1; −44.1)	(22.8; 24.3)	(11.8; 16.3)	(−50.0; −31.0)
Motor vehicle road injuries	6.3	3.3	−46.9	1.7	0.9	−47.9	4.0	2.1	−47.3
	(5.9; 6.7)	(2.8; 4.0)	(−56.5; −36.5)	(1.6; 1.8)	(0.8; 1.1)	(−56.6; −37.9)	(3.8; 4.2)	(1.8; 2.5)	(−55.9; −37.5)
Pedestrian road injuries	25.7	8.7	−66.4	8.2	2.5	−68.9	16.7	5.4	−67.8
	(24.8; 26.7)	(7.2; 10.3)	(−71.8; −59.8)	(7.7; 8.7)	(2.2; 3.0)	(−74.2; −63.4)	(16.1; 17.3)	(4.6; 6.3)	(−72.8; −62.1)
Motorcyclist road injuries	3.5	9.9	183.3	0.6	1.5	136.4	2.0	5.6	178.4
	(3.2; 3.7)	(8.2; 11.6)	(136.6; 235.5)	(0.6; 0.7)	(1.2; 1.7)	(95.5; 183.3)	(1.9; 2.1)	(4.7; 6.5)	(133.8; 226.3)
Cyclist road injuries	1.1	1.4	29.9	0.1	0.1	−12.4	0.6	0.7	22.1
	(1.0; 1.2)	(1.2; 1.7)	(5.2; 57.1)	(0.1; 0.2)	(0.1; 0.2)	(−28.7; −8.0)	(0.6; 0.7)	(0.6; 0.9)	(−0.2; 45.3)
Other road injuries	0.4	0.3	−27.7	0.1	0.1	−30.7	0.2	0.2	−29.3
	(0.4; 0.5)	(0.2; 0.4)	(−42.9; −9.5)	(0.1; 0.1)	(0.0; 0.1)	(−44.6; −16.4)	(0.2; 0.3)	(0.1; 0.2)	(−42.4; −12.5)

Source: own elaboration based on GBD 2021.

**Table 2 ijerph-22-01201-t002:** Age-standardized YLD, YLL, and DALY rates from road injuries. Colombia, 1990 and 2021.

**Cause**	**Age-Standardized Years Lived with Disability (YLD) Rate**								
	**(95% Uncertainty Intervals)**								
	**Male**		**% Change**	**Female**		**% Change**	**Total**		**% Change**
	**1990**	**2021**	**1990–2021**	**1990**	**2021**	**1990–2021**	**1990**	**2021**	**1990–2021**
Road injuries	219.8	147.1	−29.24	83.4	46.8	−40.8	150.0	94.2	−33.8
	(161.4; 288.7)	(106.4; 194.2)	(−37.2; −29.2)	(61.2; 109.5)	(33.9; 62.7)	(−47.1; −40.8)	(110.4; 197.7)	(68.3; 124.6)	(−41.0; −33.8)
Motor vehicle road injuries	58.8	19.9	−59.2	21.5	10.0	−51.0	35.8	14.7	−57.4
	(36.1; 66.6)	(14.0; 26.6)	(−62.8; −59.2)	(15.4; 28.1)	(7.3; 13.4)	(−55.6; −51.0)	(25.4; 46.8)	(10.4; 19.7)	(−60.6; −57.4)
Pedestrian road injuries	103.2	24.7	−75.0	39.6	10.5	−71.9	70.6	17.1	−74.7
	(74.6; 137.7)	(17.6; 33.5)	(−77.2; −75.0)	(28.9; 52.8)	(7.5; 14.4)	(−75.2; −71.9)	(51.2; 94.1)	(12.2; 23.3)	(−76.9; −74.7)
Motorcyclist road injuries	20.0	70.5	273.7	7.2	15.7	136.2	13.5	41.7	229.3
	(14.4; 26.9)	(49.6; 93.9)	(230.5; 273.7)	(5.2; 9.3)	(11.2; 21.0)	(97.8; 136.2)	(9.7; 17.8)	(29.4; 55.4)	(189.5; 229.3)
Cyclist road injuries	22.4	20.8	−2.3	6.8	5.5	−13.1	14.4	12.7	−7.28979039
	(16.1; 29.9)	(14.8; 28.0)	(−12.0; −2.3)	(4.8; 9.3)	(3.9; 7.7)	(−24.5; −13.1)	(10.3; 19.4)	(9.1; 17.1)	(−16.3; −7.3)
Other road injuries	23.3	11.3	−49.6	8.3	5.1	−34.9	15.6	8.0	−46.5
	(16.7; 30.8)	(8.0; 15.1)	(−53.4; −49.6)	(5.9; 11.3)	(3.6; 7.1)	(−41.9; −34.9)	(11.2; 20.7)	(5.7; 10.8)	(−50.6; −46.5)
**Cause**	**Age-Standardized Years of Life Lost (YLL) Rate**								
	**(95% Uncertainty Intervals)**								
	**Male**		**% Change**	**Female**		**% Change**	**Total**		**% Change**
	**1990**	**2021**	**1990–2021**	**1990**	**2021**	**1990–2021**	**1990**	**2021**	**1990–2021**
Road injuries	1752.4	1110.4	−26.1	497.1	236.4	−44.2	1110.6	666.3	−30.5
	(1684.7; 1823.2)	(935.2; 1288.8)	(−46.8; −26.1)	(470.2; 525.7)	(198.2; 276.0)	(−60.1; −44.2)	(1070.8; 1152.0)	(562.7; 770.3)	(−49.5; −30.5)
Motor vehicle road injuries	328.5	169.9	−38.4	90.0	46.5	−38.2	206.3	107.3	−38.3
	(308.5; 351.1)	(142.5; 201.2)	(−57.4; −38.4)	(84.1; 96.0)	(38.7; 55.2)	(−57.0; −38.2)	(196.3; 218.2)	(90.5; 126.5)	(−56.4; −38.3)
Pedestrian road injuries	1153.3	316.5	−67.3	359.1	95.1	−68.3	747.8	201.5	−68.08
	(1101.5; 1205.4)	(266.4; 372.4)	(−77.1; −67.3)	(337.5; 380.6)	(80.0; 112.3)	(−78.1; −68.3)	(717.9; 779.4)	(170.8; 236.7)	(−77.4; −68.1)
Motorcyclist road injuries	195.5	548.7	234.6	36.0	85.4	186.6	113.6	315.9	226.6
	(181.1; 210.2)	(460.0; 642.1)	(134.2; 234.6)	(33.0; 39.1)	(71.5; 99.9)	(94.4; 186.6)	(106.4; 121.2)	(267.6; 367.6)	(133.9; 226.6)
Cyclist road injuries	54.9	62.0	35.1	7.6	6.5	5.8	30.7	33.6	30.6
	(50.5; 60.0)	(51.3; 73.4)	(−8.3; 35.1)	(6.7; 8.4)	(5.4; 7.8)	(−30.6; 5.8)	(28.5; 33.5)	(28.1; 39.7)	(−10.5; 30.6)
Other road injuries	20.2	13.3	−15.9	4.4	2.9	−17.5	12.1	8.0	−17.9
	(17.6; 23.0)	(10.8; 16.3)	(−48.5; −15.9)	(4.0; 4.9)	(2.4; 3.6)	(−47.4; −17.5)	(10.8; 13.5)	(6.6; 9.6)	(−46.7; −17.9)
**Cause**	**Age-Standardized Disability-Adjusted Life Years (DALY) Rate**								
	**(95% Uncertainty Intervals)**								
	**Male**		**% Change**	**Female**		**% Change**	**Total**		**% Change**
	**1990**	**2021**	**1990–2021**	**1990**	**2021**	**1990–2021**	**1990**	**2021**	**1990–2021**
Road injuries	1972.2	1257.5	−36.2	580.5	283.2	−51.2	1260.6	760.6	−39.7
	(1879.7; 2066.6)	(1090.6; 1455.7)	(−45.1; −27.0)	(545.4; 621.1)	(245.3; 327.2)	(−57.7; −44.1)	(1204.2; 1323.0)	(660.3; 877.9)	(−47.9; −31.3)
Motor vehicle road injuries	379.3	189.8	−50.0	111.5	56.5	−49.3	242.1	122.0	−49.6
	(354.4; 405.0)	(161.7; 223.3)	(−58.0; −41.5)	(102.9; 120.7)	(48.3; 65.9)	(−56.3; −41.4)	(227.7; 257.5)	(105.6; 141.1)	(−56.8; −41.5)
Pedestrian road injuries	1256.5	341.2	−72.8	398.7	105.5	−73.5	818.5	218.6	−73.3
	(1198.8; 1315.5)	(291.0; 397.8)	(−77.0; −68.0)	(374.8; 426.5)	(89.9; 122.8)	(−77.7; −68.8)	(781.4; 855.0)	(187.9; 253.8)	(−77.3; −68.7)
Motorcyclist road injuries	215.5	619.2	187.3	43.2	101.1	134.1	127.1	357.7	181.5
	(200.7; 232.6)	(532.0; 715.6)	(146.4; 236.1)	(39.5; 46.8)	(87.5; 117.6)	(99.2; 174.8)	(119.5; 136.0)	(308.7; 410.3)	(143.4; 224.2)
Cyclist road injuries	77.3	82.8	7.1	14.4	12.0	−16.6	45.2	46.3	2.5
	(69.2; 86.1)	(71.2; 97.5)	(−8.2; 22.5)	(12.2; 17.0)	(9.9; 14.5)	(−25.4; −6.2)	(40.4; 50.6)	(40.1; 54.4)	(−10.8; 16.8)
Other road injuries	43.5	24.6	−43.5	12.7	8.1	−36.83	27.8	16.0	−42.3
	(36.0; 51.3)	(20.4; 28.7)	(−49.8; −35.2)	(10.3; 15.8)	(6.5; 10.1)	(−42.5; −31.1)	(22.9; 33.2)	(13.3; 18.9)	(−47.7; −35.3)

Source: own elaboration based on GBD 2021.

**Table 3 ijerph-22-01201-t003:** Joinpoint analysis of the age-standardized DALY rate of road injuries by road injury category. Colombia, 1990–2021.

**Road Injuries**		**Motor Vehicle Road Injuries**		**Pedestrian Road Injuries**	
**Period**	**APC (CI)**	**Period**	**APC (CI)**	**Period**	**APC (CI)**
1990–1996	1.6 (0.6; 2.6) *	1990–1996	3.6 (1.9; 5.4) *	1990–1995	0.5 (−0.4; 1.5)
1996–2001	−7.0 (−9.2; −5.7) *	1996–2001	−6.9 (−9.8; −4.7) *	1995–2000	−9.8 (−11.2; −8.8) *
2001–2012	−1.2 (−2.2; −0.7) *	2001–2014	−3.7 (−4.4; −2.3) *	2000–2012	−4.4 (−4.9; −4.1) *
2012–2016	2.1 (0.2; 3.9) *	2014–2017	3.9 (−3.8; 5.9)	2012–2017	−1.6 (−2.6; 0.4)
2016–2021	−3.9 (−5.5; −2.8) *	2017–2021	−4.1 (−8.2; −0.9) *	2017–2021	−5.5 (−7.7; −4.3) *
**Average Annual Percent Change (AAPC) 1990–2021**					
−1.6 (−1.8; −1.5) *		−2.2 (−2.5; −1.9) *		−4.2 (−4.4; −4.1) *	
**Motorcyclist Road Injuries**		**Cyclist Road Injuries**		**Other Road Injuries**	
**Period**	**APC (CI)**	**Period**	**APC (CI)**	**Period**	**APC (CI)**
1990–1998	9.3 (7.3; 10.6) *	1990–1999	4.5 (4.1; 5.1) *	1990–1995	−0.3 (−2.3; 0.7)
1998–2002	−6.0 (−9.4; 11.0)	1999–2004	1.3 (0.2; 3.0) *	1995–1998	2.0 (−2.2; 2.8)
2002–2008	7.6 (−5.4; 12.4)	2004–2009	−3.0 (−4.0; −1.1) *	1998–2009	−1.3 (−5.5; −0.4) *
2008–2016	4.9 (0.2; 6.1) *	2009–2012	−6.5 (−7.3; −4.2) *	2009–2012	−5.6 (−6.2; −1.9) *
2016–2021	−3.9 (−6.6; −1.8) *	2012–2021	−1.0 (−1.4; −0.5) *	2012–2021	−3.1 (−3.8; −2.1) *
**Average Annual Percent Change (AAPC) 1990–2021**					
3.6 (3.3; 3.8) *		0.1 (0.0; 0.1)		−1.8 (−1.9; −1.7) *	

APC: annual percent change; CI: 95% confidence interval; * α = 0.05. Source: own elaboration based on GBD.

## Data Availability

All data employed in this manuscript are open access and are available at https://vizhub.healthdata.org/gbd-results/, (accessed on 5 February 2025).

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
