# Peer review of "Measurement of the Burden of Road Injuries in Colombia, 1990–2021"

_ijerph, 2025, doi:10.3390/ijerph22081201_

Round 1
Reviewer 1 Report
Comments and Suggestions for Authors
The manuscript titled "Measurement of the burden of road traffic injuries in Colombia, 1990-2021" assesses the burden due to road traffic injuries disaggregated by sex, age groups, and specific sub-causes. Minor revisions are recommended to improve clarity and completeness. Please find the comments below:
Method:
1. Section 2.1, the total number of cases used for interpretation should be added.
2. Descriptions, parameters, and equations specifically used in this manuscript for calculation of years of life lost (YLL), years lived with disability (YLD), and DALYs need to be added.
Results:
1. In Tables 1 and 2, adjust the numeric data in the tables. Some data has moved to the second line, making it difficult to understand.
Discussion:
1. The authors should discuss the burden due to road traffic injuries disaggregated by sex, age groups, and specific sub-causes by comparing the results in this study with results from other countries. They should discuss whether the patterns are similar or not and provide explanations for these similarities or differences.
Author Response
The manuscript titled "Measurement of the burden of road traffic injuries in Colombia, 1990-2021" assesses the burden due to road traffic injuries disaggregated by sex, age groups, and specific sub-causes. Minor revisions are recommended to improve clarity and completeness. Please find the comments below:
Method:
- Section 2.1, the total number of cases used for interpretation should be added.
We used the prevalence, mortality, and age-standardized rate estimates directly from the GBD. The data obtained are those reported in the manuscript.
- Descriptions, parameters, and equations specifically used in this manuscript for calculation of years of life lost (YLL), years lived with disability (YLD), and DALYs need to be added.
Section 2.3 describes the measures used in this study. We did not include the formulas because this specific analysis was not conducted by the authors; however, we cited the main sources that detail the calculation procedures, including the corresponding equations.
Results:
- In Tables 1 and 2, adjust the numeric data in the tables. Some data has moved to the second line, making it difficult to understand.
The file we uploaded through the journal's submission system displays correctly on our end. We are unsure why the data in Tables 1 and 2 appear misaligned.
Discussion:
- The authors should discuss the burden due to road traffic injuries disaggregated by sex, age groups, and specific sub-causes by comparing the results in this study with results from other countries. They should discuss whether the patterns are similar or not and provide explanations for these similarities or differences.
We included paragraphs addressing the findings and implications disaggregated by sex, age groups, and sub-causes.
Reviewer 2 Report
Comments and Suggestions for Authors
The manuscript adds valuable perspectives to Colombia's road safety-related epidemiology and approach, and employs sound global data and methods; overall, the article is publishable but could be improved in a few ways:
1. Figure Presentation
- Figures and abstract tables are referenced in the correct format, but inconsistent formatting in the manuscript (some figure captions are in the paragraph unlike others being separate) affects the suitable presentation.
- Visualizations, if submitted as separate files, should verify that they're suitable/neat and clear.
2. Discussion Depth
- The discussion could have more critical engagement with global comparisons beyond Mexico, and could include supported policy recommendations.
- Consider more emphasis on the possible practical implications of more and more injuries from motorcyclists.
3. Ethical Clarification
- Although the paper correctly states that no ethical clearance is required for GBD data, the manuscript would be stronger in ethical declaration with an explicit comment stating this study's ethics considerations included following the GATHER guidelines and there were no identifiable 'individual' data.
Language and Grammar
- There are little grammatical and stylistic mistakes (likely from translation) that are somewhat frequent, e.g., awkward phrases like “reflected a reduction of -1.8%” could suggest they show “a 1.8 % reduction.”
- Some elements are dummied-up sections with repetitive sentence structures; the authors should consider revising to make them clearer and have better readability.
Author Response
The manuscript adds valuable perspectives to Colombia's road safety-related epidemiology and approach, and employs sound global data and methods; overall, the article is publishable but could be improved in a few ways:
- Figure Presentation
- Figures and abstract tables are referenced in the correct format, but inconsistent formatting in the manuscript (some figure captions are in the paragraph unlike others being separate) affects the suitable presentation.
We added the Excel tables to improve readability.
- Visualizations, if submitted as separate files, should verify that they're suitable/neat and clear.
The figures were incorporated directly into the manuscript file.
- Discussion Depth
- The discussion could have more critical engagement with global comparisons beyond Mexico, and could include supported policy recommendations.
We included a comparison in the discussion with countries in Latin America regarding mortality patterns from transport injuries.
- Consider more emphasis on the possible practical implications of more and more injuries from motorcyclists.
We placed greater emphasis on injuries from motorcyclists.
- Ethical Clarification
- Although the paper correctly states that no ethical clearance is required for GBD data, the manuscript would be stronger in ethical declaration with an explicit comment stating this study's ethics considerations included following the GATHER guidelines and there were no identifiable 'individual' data.
Both Section 2.1 and the “Informed Consent Statement” specify that this observational study is based on data from the GBD and that the analysis followed the GBD Protocol, which adheres to the GATHER guidelines.
Comments on the Quality of English Language
Language and Grammar
- There are little grammatical and stylistic mistakes (likely from translation) that are somewhat frequent, e.g., awkward phrases like “reflected a reduction of -1.8%” could suggest they show “a 1.8 % reduction.”
We made the correction
- Some elements are dummied-up sections with repetitive sentence structures; the authors should consider revising to make them clearer and have better readability.
The entire document has been reviewed to enhance clarity.
Reviewer 3 Report
Comments and Suggestions for Authors
Title:
Measurement of the burden of road traffic injuries in Colombia, 1990-2021
Reviewer’s comments
The manuscript needs editing to improve clarity.
This is a study of unintentional injuries due to transportation (ICD-10 codes V01 through V99), including injuries occurring on land but off-road, on water, on rail, or in the air), and injuries from non-motor vehicles (i.e., pedal cycles, animals or animal-drawn carts, boats, trains, or aircraft). The terms “road traffic injury” (or injuries) and “road injury” or (injuries) are inadequate and misleading, therefore replace them with “transportation injury” (or injuries) in the title and throughout the text and tables.
The term “cyclist” is ambiguous: does it exclude motorcyclists? Replace “cyclist” with “pedal cyclist” throughout the text and tables.
“UI” is ambiguous because it does not specify the degree of uncertainty. Replace “UI” with (“95% UI”) throughout the text and tables.
“YLD” does not mean “years lived with disability”; this is conceptually incorrect. YLD means years lost due to disability. You need to correct this throughout the text, and in the explanation of how YLD is calculated (see below).
Tables 1 and 2 are poorly formatted and difficult to read. Reduce the number of decimal places: round to the nearest whole number, or one decimal place if the number is less than 10. Increase the column widths so that the 95% uncertainty intervals remain on the same row. Demarcate injury categories with an empty row or a horizontal line. I advise that you divide each of these tables into three parts, and put each part on its own page, in landscape orientation. After you improve the readability of these tables I may have further comments on the appropriateness of the accompanying text and the discussion.
Abstract
P1, L17: Replace “years lived with disability” with “years lost due to disability”.
2.1 Source of information
P2, L78: Replace “years lived with disability” with “years lost due to disability”.
2.2 Definition and classification
P2, L84: Replace “due to road traffic injuries” with “of transportation injuries”.
P2, L85: Replace “road traffic injury” with “transportation injury”.
P2, L86: Replace “bicycle” with “pedal cycle”.
P2, L88: Replace “road traffic injuries” with “transportation injuries”.
P2, L89: Replace “road traffic injuries” with “transportation injuries”.
P2, L89: Replace “further divided into five subgroups” with “classified into five comprehensive and mutually exclusive categories”.
P3, L90: Replace “run over” with “pedestrians”.
P3, L90: Replace “cyclists” with “pedal cyclists”.
P3, L90-92: Specify the ICD-10 codes that comprise each of the five categories. In particular, you need to clarify which category includes three-wheel (V30-39) and four-wheel all-terrain vehicles (V86) designed primarily for off-road use. Such vehicles may be part of the explanation for the remarkable increase in “motorcyclist” injuries, as in North America.
2.3 Measurements
P3, L95: Replace “years lived with disability” with “years lost due to disability”.
P3, L95: Replace “DALY” with “disability-adjusted life-years lost (DALY)”.
P3, L96: Replace “sub-causes (road users)” with “categories of transportation injuries”.
P3, L106: Replace “adjusted” with “standardized”. After “applying” insert “the age- and sex-specific incidence rates in Colombia to”.
P3, L113-115: Beginning with “In turn, YLD…” replace the last three sentences of the paragraph with, “YLD was calculated by multiplying the number of prevalent non-fatal injuries in each population cell defined by age, sex and year by the Disability Weight (0≤ DW ≤1) for the specific injury. DW represents the imputed average functional loss due to the injury (estimated from previous surveys by the GBD), where the lower limit of the scale (DW=0) means no functional loss, and the upper limit (DW=1) means loss of all function (i.e., complete disability, practically equivalent to death). For example, one person living for one year with DW=0.2 would be considered to have lost 0.2 years due to disability (YLD=0.2). Aggregated total YLD is calculated as the sum of the cell-specific YLDs. DALY is calculated as the sum of YLL and YLD [13,14,19].”
3.1. Prevalence, incidence, and mortality due to road traffic injuries
P3,L135-137: The stated “prevalence” numbers do not make sense. If these are absolute counts of prevalent cases, how can they be fractional? If these are estimated standardized numbers of prevalent cases then say so, and round to the nearest whole number. If these are prevalence rates, what is the denominator? Prevalence rates are typically stated as cases per 1,000 population, in which case a prevalence of 583,166.2 seems impossibly large. Correct the stated prevalence numbers and clarify what you mean.
P3,L138: How can the number of incident cases be fractional? (See above comments about prevalence reporting). Correct the stated incidence numbers and clarify what you mean.
P4,L144-156: The repetition of results shown in Table 1 is not necessary. Replace it with a short summary, to the effect of, “Table 1 shows that age-standardized prevalence, incidence and mortality declined substantially between 1990 and 2021, for both males and females, in most injury categories. Exceptions to the pattern occurred in the category of motorcyclist injuries, where we observed substantial increase of prevalence, incidence and mortality among both males and females, and in the category of pedal cyclist injuries, where we observed statistically significant increase of mortality among males.”
Table 1
Delete the words “sub-causes and percentage change” from the table title.
P6, L169: Replace “inhabitants” with “person-years”.
Table 2
In the table title, replace “years lived with disability” with “years lost due to disability”, and after “rate” insert “per 100,000 person-years”.
4.1 Study limitations
Comment on the quality of information regarding incidence and prevalence of non-fatal injuries.
Abbreviations
Replace “years lived with disability” with “years lost due to disability”.
Author Response
Title:
Measurement of the burden of road traffic injuries in Colombia, 1990-2021
Reviewer’s comments
The manuscript needs editing to improve clarity.
The entire document has been reviewed to enhance clarity.
This is a study of unintentional injuries due to transportation (ICD-10 codes V01 through V99), including injuries occurring on land but off-road, on water, on rail, or in the air), and injuries from non-motor vehicles (i.e., pedal cycles, animals or animal-drawn carts, boats, trains, or aircraft). The terms “road traffic injury” (or injuries) and “road injury” or (injuries) are inadequate and misleading, therefore replace them with “transportation injury” (or injuries) in the title and throughout the text and tables.
We revised the title to: “Measurement of the burden of transport injuries in Colombia, 1990–2021”. The term “transportation injuries” is used throughout the manuscript.
The term “cyclist” is ambiguous: does it exclude motorcyclists? Replace “cyclist” with “pedal cyclist” throughout the text and tables.
We used the GBD term “cyclist road injuries” and “pedal cyclist” where appropriate.
“UI” is ambiguous because it does not specify the degree of uncertainty. Replace “UI” with (“95% UI”) throughout the text and tables.
95% UI” was consistently used throughout the manuscript and tables.
“YLD” does not mean “years lived with disability”; this is conceptually incorrect. YLD means years lost due to disability. You need to correct this throughout the text, and in the explanation of how YLD is calculated (see below).
We decided to retain the term as defined by the GBD Study, where YLD stands for “Years Lived with Disability” (see: https://www.healthdata.org/research-analysis/about-gbd)
Tables 1 and 2 are poorly formatted and difficult to read. Reduce the number of decimal places: round to the nearest whole number, or one decimal place if the number is less than 10. Increase the column widths so that the 95% uncertainty intervals remain on the same row. Demarcate injury categories with an empty row or a horizontal line. I advise that you divide each of these tables into three parts, and put each part on its own page, in landscape orientation. After you improve the readability of these tables I may have further comments on the appropriateness of the accompanying text and the discussion.
All figures in the manuscript are reported with one decimal place, in both text and tables/figures; thus, we cannot reduce them further. The version we submitted includes Tables 1 and 2 in landscape orientation, and the values are displayed correctly. We are unsure why they are not displaying properly on the reviewer’s end.
Abstract
P1, L17: Replace “years lived with disability” with “years lost due to disability”.
We decided to retain the term as defined by the GBD Study, where YLD stands for “Years Lived with Disability” (see: https://www.healthdata.org/research-analysis/about-gbd)
2.1 Source of information
P2, L78: Replace “years lived with disability” with “years lost due to disability”.
We decided to retain the term as defined by the GBD Study, where YLD stands for “Years Lived with Disability” (see: https://www.healthdata.org/research-analysis/about-gbd)
2.2 Definition and classification
P2, L84: Replace “due to road traffic injuries” with “of transportation injuries”.
We corrected the term.
P2, L85: Replace “road traffic injury” with “transportation injury”.
We corrected the term.
P2, L86: Replace “bicycle” with “pedal cycle”.
We corrected the term.
P2, L88: Replace “road traffic injuries” with “transportation injuries”.
We corrected the term.
P2, L89: Replace “road traffic injuries” with “transportation injuries”.
We corrected the term.
P2, L89: Replace “further divided into five subgroups” with “classified into five comprehensive and mutually exclusive categories”.
We corrected the term.
P3, L90: Replace “run over” with “pedestrians”.
We corrected the term.
P3, L90: Replace “cyclists” with “pedal cyclists”.
We corrected the term.
P3, L90-92: Specify the ICD-10 codes that comprise each of the five categories. In particular, you need to clarify which category includes three-wheel (V30-39) and four-wheel all-terrain vehicles (V86) designed primarily for off-road use. Such vehicles may be part of the explanation for the remarkable increase in “motorcyclist” injuries, as in North America.
We included the ICD-10 codes corresponding to each of the five injury categories analyzed.
2.3 Measurements
P3, L95: Replace “years lived with disability” with “years lost due to disability”.
We decided to retain the term as defined by the GBD Study, where YLD stands for “Years Lived with Disability” (see: https://www.healthdata.org/research-analysis/about-gbd)
P3, L95: Replace “DALY” with “disability-adjusted life-years lost (DALY)”.
We corrected the term.
P3, L96: Replace “sub-causes (road users)” with “categories of transportation injuries”.
We corrected the term.
P3, L106: Replace “adjusted” with “standardized”. After “applying” insert “the age- and sex-specific incidence rates in Colombia to”.
We made the correction from “adjusted” to “standardized”; however, we did not include the phrase “the age- and sex-specific incidence rates in Colombia to” because it does not reflect what was done. In the GBD Study, mortality rates were standardized, as we specify in the paragraph.
P3, L113-115: Beginning with “In turn, YLD…” replace the last three sentences of the paragraph with, “YLD was calculated by multiplying the number of prevalent non-fatal injuries in each population cell defined by age, sex and year by the Disability Weight (0≤ DW ≤1) for the specific injury. DW represents the imputed average functional loss due to the injury (estimated from previous surveys by the GBD), where the lower limit of the scale (DW=0) means no functional loss, and the upper limit (DW=1) means loss of all function (i.e., complete disability, practically equivalent to death). For example, one person living for one year with DW=0.2 would be considered to have lost 0.2 years due to disability (YLD=0.2). Aggregated total YLD is calculated as the sum of the cell-specific YLDs. DALY is calculated as the sum of YLL and YLD [13,14,19].”
We appreciate the suggestion; however, we were unable to locate the source cited by the reviewer and want to ensure that what is described accurately reflects the methodology used in the GBD Study.
3.1. Prevalence, incidence, and mortality due to road traffic injuries
P3,L135-137: The stated “prevalence” numbers do not make sense. If these are absolute counts of prevalent cases, how can they be fractional? If these are estimated standardized numbers of prevalent cases then say so, and round to the nearest whole number. If these are prevalence rates, what is the denominator? Prevalence rates are typically stated as cases per 1,000 population, in which case a prevalence of 583,166.2 seems impossibly large. Correct the stated prevalence numbers and clarify what you mean.
As specified in the manuscript, the GBD does not rely solely on raw case counts but rather on Bayesian statistical models to adjust, impute, and extrapolate data from multiple sources (surveys, records, censuses, literature, etc.). As a result, the reported figures are modeled averages or expected rates, not whole numbers.
P3,L138: How can the number of incident cases be fractional? (See above comments about prevalence reporting). Correct the stated incidence numbers and clarify what you mean.
As specified in the manuscript, the GBD does not rely solely on raw case counts but rather on Bayesian statistical models to adjust, impute, and extrapolate data from multiple sources (surveys, records, censuses, literature, etc.). As a result, the reported figures are modeled averages or expected rates, not whole numbers.
P4,L144-156: The repetition of results shown in Table 1 is not necessary. Replace it with a short summary, to the effect of, “Table 1 shows that age-standardized prevalence, incidence and mortality declined substantially between 1990 and 2021, for both males and females, in most injury categories. Exceptions to the pattern occurred in the category of motorcyclist injuries, where we observed substantial increase of prevalence, incidence and mortality among both males and females, and in the category of pedal cyclist injuries, where we observed statistically significant increase of mortality among males.”
We revised the paragraph explaining Table 1 based on the reviewer’s suggestion.
Table 1
Delete the words “sub-causes and percentage change” from the table title.
We corrected the term.
P6, L169: Replace “inhabitants” with “person-years”.
We did not apply this change because we are reporting standardized rates per 100,000 people for each indicator. If the reported figures were total years of YLL, YLD, or DALY, the appropriate term would be “person-years”.
Table 2
In the table title, replace “years lived with disability” with “years lost due to disability”, and after “rate” insert “per 100,000 person-years”.
We did not apply this change because we are reporting standardized rates per 100,000 people for each indicator. If the reported figures were total years of YLL, YLD, or DALY, the appropriate term would be “person-years”.
4.1 Study limitations
Comment on the quality of information regarding incidence and prevalence of non-fatal injuries.
We included information regarding this point in the study’s limitations section.
Abbreviations
Replace “years lived with disability” with “years lost due to disability”.
We decided to retain the term as defined by the GBD Study, where YLD stands for “Years Lived with Disability” (see: https://www.healthdata.org/research-analysis/about-gbd)
Round 2
Reviewer 3 Report
Comments and Suggestions for Authors
Title:
Measurement of the burden of transport injuries in Colombia, 1990-2021
Reviewer’s comments
Editing for clarity is still needed. Clear definitions of the injury categories are of critical importance.
2.2 Definition and classification
P2, L89: Delete the words “comprehensive and”.
P2, L89-93: The stated definitions of the injury categories do not make sense, because the category names do not agree with the stated ICD-10 codes. Verify and correct these, otherwise your readers will not understand what injuries you are describing (I certainly do not, from what is written). For your information, these are the ranges of ICD-10 V-codes for external causes:
V00-V09 Pedestrian injured in transport accident
V10-V19 Pedal cycle rider injured in transport accident
V20-V29 Motorcycle rider injured in transport accident
V30-V39 Occupant of three-wheeled motor vehicle injured in transport accident
V40-V49 Car occupant injured in transport accident
V50-V59 Occupant of pick-up truck or van injured in transport accident
V60-V69 Occupant of heavy transport vehicle injured in transport accident
V70-V79 Bus occupant injured in transport accident
V80-V89 Other land transport accidents
V80 Animal-rider or occupant of animal-drawn vehicle injured in transport accident
V81 Occupant of railway train or railway vehicle injured in transport accident
V82 Occupant of powered streetcar injured in transport accident
V83 Occupant of special vehicle mainly used on industrial premises injured in transport accident
V84 Occupant of special vehicle mainly used in agriculture injured in transport accident
V85 Occupant of special construction vehicle injured in transport accident
V86 Occupant of special all-terrain or other off-road motor vehicle, injured in transport accident
V87 Traffic accident of specified type but victim's mode of transport unknown
V88 Nontraffic accident of specified type but victim's mode of transport unknown
V89 Motor- or nonmotor-vehicle accident, type of vehicle unspecified
V90-V94 Water transport accidents
V95-V97 Air and space transport accidents
V98 Other specified transport accidents
V99 Unspecified transport accident
3.1. Prevalence, incidence, and mortality of transport injuries
P3,L135-137: Is the stated “prevalence” an expected number of cases or a prevalence rate? If you mean a prevalence rate, what is the denominator? Prevalence rates are typically stated as cases per 1,000 population, in which case a prevalence of 583,166.2 seems impossibly large. Clarify this for your readers.
Author Response
Reviewer’s comments
Editing for clarity is still needed. Clear definitions of the injury categories are of critical importance.
2.2 Definition and classification
P2, L89: Delete the words “comprehensive and”.
The words "comprehensive and" were removed. We included them because one of the reviewers suggested them.
P2, L89-93: The stated definitions of the injury categories do not make sense, because the category names do not agree with the stated ICD-10 codes. Verify and correct these, otherwise your readers will not understand what injuries you are describing (I certainly do not, from what is written). For your information, these are the ranges of ICD-10 V-codes for external causes:
V00-V09 Pedestrian injured in transport accident
V10-V19 Pedal cycle rider injured in transport accident
V20-V29 Motorcycle rider injured in transport accident
V30-V39 Occupant of three-wheeled motor vehicle injured in transport accident
V40-V49 Car occupant injured in transport accident
V50-V59 Occupant of pick-up truck or van injured in transport accident
V60-V69 Occupant of heavy transport vehicle injured in transport accident
V70-V79 Bus occupant injured in transport accident
V80-V89 Other land transport accidents
V80 Animal-rider or occupant of animal-drawn vehicle injured in transport accident
V81 Occupant of railway train or railway vehicle injured in transport accident
V82 Occupant of powered streetcar injured in transport accident
V83 Occupant of special vehicle mainly used on industrial premises injured in transport accident
V84 Occupant of special vehicle mainly used in agriculture injured in transport accident
V85 Occupant of special construction vehicle injured in transport accident
V86 Occupant of special all-terrain or other off-road motor vehicle, injured in transport accident
V87 Traffic accident of specified type but victim's mode of transport unknown
V88 Nontraffic accident of specified type but victim's mode of transport unknown
V89 Motor- or nonmotor-vehicle accident, type of vehicle unspecified
V90-V94 Water transport accidents
V95-V97 Air and space transport accidents
V98 Other specified transport accidents
V99 Unspecified transport accident
This study focuses exclusively on road injuries, as defined by the GBD framework. This category includes injuries from land transport incidents on public roads and excludes non-road or non-land transport injuries. This approach ensures alignment with GBD definitions and facilitates targeted analysis of road traffic injuries (ICD-10: V01–V04.9, V06–V80.9, V82–V82.9, V87.2–V87.3), excluding, for instance, codes related to water, air, or other unspecified transport (e.g., V90–V98.8). This distinction ensures that the analysis is specific to injuries occurring on public roads and highways, in line with global efforts to monitor road traffic safety and related health outcomes. See the GBD ICD-10 codes at:
- https://ghdx.healthdata.org/record/ihme-data/gbd-2021-cause-icd-code-mappings
- https://www.healthdata.org/research-analysis/diseases-injuries-risks/factsheets/2021-road-injuries-level-3-disease
We have also updated the references to these selected codes and revised all instances of “transport injuries” to “road injuries” throughout the manuscript
3.1. Prevalence, incidence, and mortality of transport injuries
P3,L135-137: Is the stated “prevalence” an expected number of cases or a prevalence rate? If you mean a prevalence rate, what is the denominator? Prevalence rates are typically stated as cases per 1,000 population, in which case a prevalence of 583,166.2 seems impossibly large. Clarify this for your readers
We have specified that the first paragraph refers to the number of prevalent cases, whereas Table 1 presents the age-standardized prevalence, incidence, and mortality rates (per 100,000 inhabitants).

Round 3
Reviewer 3 Report
Comments and Suggestions for Authors
Title:
Measurement of the burden of road injuries in Colombia, 1990-2021
Reviewer’s comments
The authors have still not sufficiently addressed my comments regarding the need for clear definitions of the injury categories. As I stated before, this is of critical importance. If they do not address the matter then the article is not fit for publication.
2.2 Definition and classification
P2, L93-97: Assuming that the stated ICD-10 codes are correct, then Category 1 (“motor vehicle”, sic) should be called “pedestrian road injuries”. Category 2 (“pedestrians”, sic) should be called “cyclist road injuries”. Category 4 (“cyclists”, sic) should be called “motor vehicle road injuries”. Replace the verbal description of Category 5 (“other”, sic) with “5) other road injuries (animal riders, occupants of animal-drawn vehicles, and occupants of streetcars)”.
For your information, according to your reference (https://ghdx.healthdata.org/record/ihme-data/gbd-2021-cause-icd-code-mappings) these are the ranges of ICD-10 V-codes for GBD categories of Road injuries (V01-V04.99, V06-V80.929, V82-V82.9, V87.2-V87.3):
Pedestrian road injuries V01-V04.99, V06-V09.9
Cyclist road injuries V10-V19.9
Motorcyclist road injuries V20-V29.9
Motor vehicle road injuries V30-V79.9, V87.2-V87.3
Other road injuries V80-V80.929, V82-V82.9
Author Response
We thank the reviewer for pointing out this error. We have corrected the ICD-10 codes associated with road injuries. The following adjustment was made:
"For this study, road injuries were classified into five mutually exclusive categories of road injuries: 1) pedestrian road injuries (V01-V04.9, V06-V09.9); 2) cyclist road injuries (V10-V19.9); 3) motorcyclist road injuries (V20-V29.9); 4) motor vehicle road injuries (V30-V79.9, V87.2-V87.3); 5) other road injuries (for example, water transport injuries, air transport injuries and those that are unspecified) (V80-V80.9, V82-V82.9) [16-18]".

Round 4
Reviewer 3 Report
Comments and Suggestions for Authors
Editing is still needed before publication.
2.2 Definition and classification
P3, L96-97: Delete the incorrect statement “5) other road injuries (for example, water transport injuries, air transport injuries and those that are unspecified): and replace it with “5) other road injuries (animal riders, occupants of animal-drawn vehicles, and occupants of streetcars)”.
Author Response
We thank the reviewer for their valuable suggestions. In response, we have made the following adjustment to the text: “Other road injuries (e.g., animal riders, occupants of animal-drawn vehicles, and occupants of agricultural or special vehicles).”